# Temperature fluctuation alleviates the negative effects of warming on marine diatoms: comparison between *Thalassiosira* sp. and *Nitzschia closterium* f. minutissima

Yangjie Sheng<sup>1#</sup>, Yanan Wang<sup>3#</sup>, Ting Cai<sup>4</sup>, Yuntao Wang<sup>5</sup>, Afef Fathalli<sup>6</sup>, Sana Ben Ismail<sup>6</sup>, Yuanyuan Feng<sup>1,2\*</sup>

<sup>1</sup>State Key Laboratory of Submarine Geoscience; Key Laboratory of Polar Ecosystem and Climate Change, Ministry of Education; Shanghai Key Laboratory of Polar Life and Environment Sciences; and School of Oceanography, Shanghai Jiao Tong University, 1954 Huashan Road, Shanghai 200030, China.

<sup>3</sup>Shenyang Wanling Biotechnology Co., Ltd, Liaoning, China.

15

#Yangjie Sheng and Yanan Wang contributed equally to this work

Abstract. Marine phytoplankton are subjected to a wide range of environmental heterogeneity from mean climate change to natural fluctuations under the climate change scenario. These changes include the changes in the frequency of temperature fluctuations of the sea surface. Here we conducted semi-continuous incubation experiments on two ecologically significant marine diatom species, *Thalassiosira* sp. and *Nitzschia closterium* f. minutissima, to examine the physiological responses to ocean warming and temperature fluctuation (± 4 °C) under low (20 °C) and high (25 °C) average temperatures. Our results demonstrate that temperature fluctuation alleviated the negative effects of elevated temperatures on the growth of both species. For *Thalassiosira* sp., warming under constant temperature significantly reduced the growth rate, but significantly increased the cellular elemental contents, and sinking rate. However, warming significantly reduced cellular particulate organic carbon (POC) and biogenic silica (BSi) contents, and sinking rate, while increasing the protein content to cope with the thermal stress under temperature fluctuation. The effects of temperature fluctuation were dependant on the average temperature: at low average temperatures, temperature fluctuation increased cellular POC, BSi, POC productivity, and sinking rates, whereas at high average temperatures, these parameters were decreased significantly. For Nitzschia closterium f. minutissima, warming under both constant and fluctuated temperatures significantly increased the POC, particulate organic nitrogen (PON) and POP quotas. The interaction between warming and temperature fluctuation had antagonistic effects on most parameters examined for *Thalassiosira* sp.; whereas had synergistic effects on the physiological parameters of Nitzschia closterium f. minutissima. Overall, Nitzschia closterium f. minutissima exhibited stronger tolerance to warming and temperature fluctuation, suggesting species-specific responses of diatoms to warming and temperature fluctuations.

<sup>&</sup>lt;sup>2</sup>Laboratory for Polar Science, Polar Research Institute of China, Ministry of Natural Resources, Shanghai 200136, China.

<sup>&</sup>lt;sup>4</sup>WuXi AppTec Co., Ltd, Shanghai, China.

<sup>&</sup>lt;sup>5</sup>State Key Laboratory of Satellite Ocean Environment Dynamics, Second Institute of Oceanography, Ministry of Natural Resources, Hangzhou, Zhejiang, China. <sup>6</sup>Shenyang Wanling Biotechnology Co., Ltd, Liaoning, China.

<sup>&</sup>lt;sup>6</sup>Laboratory Milieu Marin, Institute National des Sciences et Technologies de la Mer, Tunis, Tunisie.

<sup>\*</sup>Correspondence to: Yuanyuan Feng (yuanyuan.feng@sjtu.edu.cn)

These findings highlight the important, yet often underestimated, influence of temperature fluctuation on the physiology of marine diatoms in the context of global warming, thus having implications for further understanding the biogeochemical feedbacks.

#### 1 Introduction

40

45

The continuous anthropogenic emissions of carbon dioxide (CO<sub>2</sub>) have led to a steady increase in atmospheric CO<sub>2</sub> concentrations (IPCC, 2023). This will in turn induce global warming, with an average increase in global temperature by approximately 1 °C above 1850–1900 in 2011–2020. (IPCC, 2023). Under current high-emission scenarios, it is projected that the global sea surface temperature will increase by at least 2-4 °C by the end of the 21st century (Bindoff et al., 2019). Temperature is a key factor affecting enzyme activity and plays a crucial role in regulating the metabolic rates of phytoplankton (Eisenthal et al., 2006; Eppley, 1972; Falkowski and Oliver, 2007). According to model projections, the net primary productivity of marine phytoplankton may decline by around 20 % during the 21st century as a result of ocean warming (Steinacher et al., 2010). Within a certain temperature range, the growth rate, photosynthesis, respiration, and other key physiological processes of phytoplankton increase with rising temperature (Barton et al., 2020), which in turn affects the utilization of nutrients within cells (Raven and Geider, 2006). Moreover, warming favors smaller-sized phytoplankton, as these cells can more effectively acquire nutrients and utilize CO<sub>2</sub> (Zaoli et al., 2019; MorÁn et al., 2010). Therefore, small-sized phytoplankton typically demonstrate stronger responses to warming in terms of both growth rates and photosynthetic activity (Wang et al., 2024), affecting the size fractionation of marine primary producers and thus the carbon sequestration (Anderson et al., 2021).

In addition, marine phytoplankton are also subjected to environmental heterogeneity in the ocean (Bernhardt et al., 2020). As climate change accelerates, the frequency and amplitude of marine environmental fluctuations are also expected to change (Perkins-Kirkpatrick and Lewis, 2020). Ocean warming not only increases the average ocean temperature but also enhances the frequency and intensity of temperature fluctuation, which may which may have complex effects on marine organisms than warming alone (Ketola and Saarinen, 2015). It has been reported that high-frequency temperature fluctuation (2 days) reduced the mortality rate of *Emiliania huxleyi*, a dominant species of the calcifying phytoplankton group that play a key role in the calcium carbonate production and the marine carbon cycling (Wang et al., 2019). For marine diazotroph *Trichodesmium*, temperature fluctuation was found to reduce the growth rate under phosphorus-replete conditions (Qu et al., 2019). Additionally, study on green algae has shown that temperature fluctuation slowed the growth rate of *Chlorella* and *Micromonas*, but did not affect the growth of *Microcystis aeruginosa* (Zhang et al., 2015). Furthermore, temperature fluctuation may enhance the ability of diatom *Thalassiosira pseudonana* to better adapt to high temperature (Schaum, 2018; Schaum et al., 2018). Therefore, different phytoplankton groups exhibit species-specific responses to temperature fluctuation, and temperature fluctuation may alter the taxonomic composition of phytoplankton community. A mesoscale enclosure experiment conducted in the coastal water of southern China found that temperature fluctuation reduced the abundance of

diatoms, while increasing the proportion of *Synechococcus* and *Prasinophytes*, which may consequently influence the upper trophic levels in the marine food web (Zhang et al., 2025).

Marine diatoms, a critical phytoplankton functional group, contribute approximately 40 % of the ocean's primary productivity and play a key role in biogeochemical cycles as a central element of the marine biological carbon pump (Qiao et al., 2021; Sanders et al., 2014; Field et al., 1998). The growth responses to temperature variations are delineated by thermal tolerance curves (Thomas et al., 2012). An increase in temperature promotes growth until the optimal temperature is reached, beyond which further elevation in temperature results in a decrease in growth rate and even cease of growth (Edullantes et al., 2023).

Different diatom species have distinct optimal growth temperatures. For instance, *Paralia suicata* demonstrates optimal growth at 6.25 °C (He et al., 2024), while *Thalassiosira weissflogii* thrives at 19.1 °C (Rossi et al., 2023); *Thalassiosira pseudonana* and *Nitzschia frustulum* exhibit optimal growth at 20.9 °C (Kuefner et al., 2020) and 28.5 °C (Rossi et al., 2023), respectively. As such, the response of diatoms to warming may be species specific. For instance, 5 °C of warming significantly reduced the biomass of *Phaeodactylum tricornutum* while having no significant effect on *Thalassiosira* weissflogii (Zeng et al., 2020). 4 °C of warming significantly reduced the particulate organic carbon content of *Thalassiosira* sp., whereas the rate of particulate organic carbon production increased with temperature for *Nitzschia closterium* f.minutissima (Cai et al., 2022). The effects of warming also vary with the duration of exposure; short-term warming has little effect on the growth rate of *P. tricornutum*, while long-term exposure significantly inhibits the growth (Hong et al., 2023).

Although the impacts of warming on marine diatoms have been extensively investigated, our understanding of how temperature fluctuations affect the physiological responses of different diatom species remains limited. This knowledge gap represents a critical constraint in developing a comprehensive understanding of marine diatoms' adaptive responses to environmental changes under more ecologically realistic scenarios. In this study, we investigated the individual and interactive effects of warming and temperature fluctuations on the physiology of marine diatoms by conducting semi-continuous incubation experiments. Two representative diatom species were selected: the centric diatom *Thalassiosira* sp. and the pennate diatom *Nitzschia closterium* f. minutissima. Both species are model diatoms, representing the two major taxonomic class of diatoms (Centric vs. Pennate), allowing for comparison of responses to environmental changes between two distinct diatom groups. To assess their physiological responses and potential implications for the marine carbon cycle considering both warming and temperature fluctuation, we systematically measured the key parameters including growth rate, macromolecular composition, elemental stoichiometry, and sinking rate.

# 2 Materials and Methods

# 2.1 Algae culture

The marine diatoms *Thalassiosira* sp. and *Nitzschia closterium* f. minutissima were originally isolated from surface seawater (depth 3–12 m; salinity 34.78) at 118°58.055′E, 38°39.111′N (China) in August 2019. At the time of sampling, the sea surface temperature was approximately 15 °C. Stock cultures of two diatoms were maintained in sterilized f/2 medium natural seawater, in Nalgene polycarbonate bottles in a constant temperature and illumination incubator (GXZ-280D, NingBo, China) at 20 °C.

## 2.2 Experimental setup

Four temperature treatments were established (Fig. 1, Table 1): (i) LTCT: constant 20 °C; HTCT: constant 25 °C; (iii) LTFT: 105 fluctuated between 16 °C and 24 °C per day (20 ± 4 °C); (iv) HTFT: a fluctuated based on warming, which cycled between 21 and 29 °C per day (25 ± 4 °C). The selection of 20 °C and 25 °C was based on both ecological relevance and experimental design considerations. We have previously obtained the thermal response curves of the two species, the optimal growth temperature for *Thalassiosira* sp. was ~ 19 °C, and ~ 22 °C for *Nitzschia closterium* f. minutissima (Cai et al., 2022). The temperature condition in each treatment was maintained in a water tank, connected to a seawater temperature controller (HC150A, Hailea, China). During the course of incubation, temperature in the water tanks were monitored continuously using a HOBO underwater temperature logger (MX2201, USA). The measured temperature was within a range of ±0.2 °C from the preset temperature.

For each temperature treatment, stock cultures in the logarithmic growth phase were inoculated into triplicate polycarbonate culture flasks (1L, Nalgene, USA), with an initial cell abundance of 1×10<sup>4</sup> cells mL<sup>-1</sup>. The light intensity was maintained at 170-200 µmol photons m<sup>-2</sup> s<sup>-1</sup> with light: dark cycle of 12 h: 12 h. The culture medium was prepared with seawater collected from the Yellow Sea, China, filtered using 0.2 µm membrane (Whatman, USA) with nutrients added according to the f/2 recipe. 2 mL of sample from incubation bottle were taken from the cultures every 24 hours to measure in vivo Chl *a* fluorescence using a Turner fluorometer (Trilogy, Turner Designs, USA). The cultures were diluted with freshly prepared medium to supply enough nutrients. Final sampling was carried out once the cell growth rate had stabilized, with variations of less than 10 % for over five consecutive days.

Figure 1: Temperature schematic diagram. LTCT: 20 °C, LTFT:  $20 \pm 4$  °C, HTCT: 25 °C, HTFT:  $25 \pm 4$  °C.

Table 1: Abbreviation table of temperature treatment group.

| Abbreviation | Full term                        | Temperature treatment         |  |  |
|--------------|----------------------------------|-------------------------------|--|--|
| LTCT         | Low and constant temperature     | 20 ℃                          |  |  |
| LTFT         | Low and fluctuation temperature  | $20 \pm 4~^{\circ}\mathrm{C}$ |  |  |
| HTCT         | High and constant temperature    | 25 °C                         |  |  |
| HTFT         | High and fluctuation temperature | $25 \pm 4$ °C                 |  |  |

#### 2.3 Physiological and Biogeochemical Analyses

# 2.3.1 Chlorophyll a content and growth rate

For chlorophyll *a* (Chl *a*) measurement, 20 mL sample was filtered onto GF/F filters (Whatman, USA). After extraction with 5 mL of 90 % acetone for 24 hours (stored at -20 °C in the dark), the Chl *a* content was measured using a fluorometer (Trilogy, Turner Designs, USA).

2 mL sample was kept in the dark for 20 minutes, followed by measurement of the in vivo Chl a fluorescence using a fluorometer (Trilogy, Turner Designs, USA). For the in vivo fluorescence measurements, the culture medium was always used as blank, therefore the potential interference from culture medium was eliminated. In addition, we conducted a semi-continuous incubation with cells growing at the exponential phase, and the cultures in the incubation bottles were well mixed for a 2-3 times a day, as such, there was no cell aggregation observed in the culture system. The growth rate ( $\mu$ , d<sup>-1</sup>) was then calculated using the following equation:

$$\mu = \ln \frac{N_2/N_1}{T_2 - T_1},$$

where  $N_2$  and  $N_1$  represent the Chl a fluorescence at times  $T_2$  and  $T_1$ , respectively.

#### 2.3.2 Cellular macromolecule content

The algal cultures were filtered through 0.4 μm polycarbonate filters (PALL, USA) and stored at -80 °C until the determination of protein and carbohydrate contents.

Protein content was measured using a Coomassie Brilliant Blue G-250 method (Bradford, 1976). The algal cells on the filter were eluted with 1 mL of Milli-Q water, followed by the addition of 4 mL of Coomassie Brilliant Blue solution. After thorough mixing, the mixture was left for 5 minutes to develop color. Absorbance was then measured at 595 nm using a UV-

visible spectrophotometer (UV-2550, Shimadzu, Japan).

Carbohydrate content was determined using the sulfuric acid-phenol method (Dubois et al., 1956). The filter was removed and placed in 5 mL of 0.05 mol/L H<sub>2</sub>SO<sub>4</sub>, then heated in a water bath at 60 °C for 10 minutes and cooled to room temperature. 2 mL sample was taken, to which 0.05 mL of 80 % phenol solution and 5 mL of concentrated sulfuric acid reagent were added. After standing for 30 minutes for color development, absorbance was measured at 485 nm using a UV-visible spectrophotometer (UV-2550, Shimadzu, Japan).

## 2.3.3 Elemental composition

Biogenic silica (BSi) was analyzed using a spectrophotometric (Nelson et al., 1995). Samples were filtered onto  $0.4~\mu m$  polycarbonate filters (Millipore, USA), dried in an oven at  $60~^{\circ}$ C (DPG-9420A, Shanghai), and stored in a desiccator until analysis.

Samples for particulate organic phosphorus (POP) measurement were filtered onto pre-combusted GF/F filters (450 °C for 4 hours). The filters were then rinsed with 2 mL of 0.17 M Na<sub>2</sub>SO<sub>4</sub> solution, transferred to 20 mL pre-combusted (8 hours at 450 °C) glass vials, with addition of 2 mL of 0.017 mol L<sup>-1</sup> MgSO<sub>4</sub> to each vial.

Particulate organic carbon (POC) and particulate organic nitrogen (PON) were analyzed using a CHN elemental analyzer (Costech, Italy). A 100 mL sample of algal culture was filtered onto a pre-combusted GF/F filter. The filter was then fumed with concentrated HCI for 3 hr to remove inorganic carbon and subsequently dried in a 60 °C oven. The organic carbon fixation rate (C<sub>R</sub>, pg C cell-<sup>1</sup> d-<sup>1</sup>) was calculated as:

$$C_R = POC_{cell} \times \mu$$

in which  $POC_{cell}$  is the cellular content of POC (pg C cell<sup>-1</sup>) obtained from the measurement of elemental content and  $\mu$  is the growth rate on the last day of culture.

# 165 2.3.4 Cell size and sinking rate

Cell size was measured using a laser particle size analyzer (LS 13320, Beckman, USA). Measurements were taken after the laser particle size analyzer had completed its preheating process. Each sample was measured three times.

The sinking rate was determined using the SETCOL method (Bienfang, 1981). The samples were filled into the column and stand in the dark for 3 hours. Subsequently, samples were collected from the upper, middle, and lower sections of the column.

Each sample was filtered through GF/F filters, and the Chl *a* content was measured. The sinking rates were calculated using the following equation:

$$\psi = \left(\frac{B_S}{B_t}\right) \times \frac{l}{t},$$

where  $B_t$  is the total biomass in the column ( $\mu g L^{-1}$ ),  $B_s$  is the biomass in the bottom layer of the column ( $\mu g L^{-1}$ ), l is the height of the column (cm), and t is the sinking time (hours).

## 175 3 Statistical analysis

Significance analysis and interaction effects were performed by two-way ANOVA using Origin 2021 software (OriginLab Corporation, USA). Differences between treatments were considered significant at level of p < 0.05. The pairwise tests between treatments were conducted using Tukey's multiple comparison post-hoc analysis. Three independent replicates (n = 3) were done for each experiment, and differences between treatments were considered significant at level of p < 0.05. The interaction effects for warming and temperature fluctuations were calculated as follows:

$$ME_{1+2} = (1 + OE_1) \times (1 + OE_2)-1,$$

where  $ME_{1+2}$  is the interaction effect of warming and temperature fluctuation, and  $OE_1$  and  $OE_2$  are the apparent effects of warming and temperature fluctuation on phytoplankton physiological parameters, respectively. When  $|OE_{1+2}| > |ME_{1+2}|$ , the interaction between the two environmental factors is synergistic, whereas when  $|OE_{1+2}| 

Figure 2: Growth rate of *Thalassiosira* sp. (A) and *N. closterium* f. minutissima (B) under LTCT: 20 °C, LTFT: 20 ± 4 °C, HTCT: 25 °C, HTFT: 25 ± 4 °C. The error bars represent the standard deviations (n=3) and the letters above the bars denote the statistically significant differences (p<0.05).

# 4.2 Chlorophyll a, protein and carbohydrate contents

200

205

210

215

The Chl a content of *Thalassiosira* sp. and N. *closterium* f. minutissima were significantly increased by 124.12 % and 66.44 % in HTCT compared to LTCT, respectively (p<0.05, Fig. 3A, B). Compared to LTFT, the Chl a content of N. *closterium* f. minutissima was significantly increased to 0.16  $\pm$  0.00 under HTFT (p<0.05, Fig. 3B), while no significant effect was observed on the Chl a content of *Thalassiosira* sp. (p>0.05, Fig. 3A).

At LTFT, the Chl a content of *Thalassiosira* sp. and N. *closterium* f. minutissima were significantly increased by 68.71 % and 65.78 % compared to LTCT, respectively (p<0.05, Fig. 3A, B). At HTFT, the Chl a content of *Thalassiosira* sp. was  $0.33 \pm 0.03$ , significantly decreased by 35.23 % compared to HTCT (p<0.05, Fig. 3A), while that of N. *closterium* f. minutissima was significantly increased by 144.87 % (p<0.05, Fig. 3B).

At HTFT, the cellular protein content of *Thalassiosira* sp. and *N. closterium* f. minutissima were  $1.90 \pm 0.37$  and  $1.03 \pm 0.07$ , significantly increased by 64.53 % and 115.40 % compared to LTFT, respectively (p<0.05, Fig. 3C, D). The cellular protein content of *Thalassiosira* sp. was significantly increased by 123.48 % at HTFT compared to HTCT (p<0.05, Fig. 3C), but no significant effect was observed at LTFT compared to LTCT (p>0.05, Fig. 3C). For *N. closterium* f. minutissima, temperature fluctuation significantly increased the cellular protein content by 65.78 % and 144.87 % compared to constant temperature at low and high temperature, respectively (p<0.05, Fig. 3D).

HTCT significantly increased the cellular carbohydrate content of *Thalassiosira* sp. by 103.09 % compared to LTCT, but HTFT markedly reduced it by 25.36 % compared to LTFT (p<0.05, Fig. 3E). Warming had no significant effect on the cellular carbohydrate content of *N. closterium* f. minutissima at either constant temperature or under temperature fluctuation (p>0.05, Fig. 3F). At LTFT, the cellular carbohydrate content of *Thalassiosira* sp. was  $8.31 \pm 0.32$ , significantly increased

by 80.25 % compared to LTCT, but the carbohydrate content of *Thalassiosira* sp. was  $1.82 \pm 0.28$ , decreased by 33.76 % at HTFT compared to HTCT (p

Figure 3: The cellular contents of Chl a (A, B), protein (C, D), and carbohydrate (E, F) of *Thalassiosira* sp. (A, C, E) and N. closterium f. minutissima (B, D, F) under LTCT: 20 °C, LTFT: 20 ± 4 °C, HTCT: 25 °C, HTFT: 25 ± 4 °C. The error bars represent the standard deviations (n=3) and the letters above the bars denote the statistically significant differences (p

Figure 4: Cellular contents (pg cell<sup>-1</sup>) of particulate organic carbon (POC, A, B), particulate organic nitrogen (PON, C, D), particulate organic phosphorus (POP, E, F), biogenic silica (BSi, G, H) for *Thalassiosira* sp. (A, C, E, G) and *N. closterium* f. minutissima (B, D, F, H) grown under LTCT: 20 °C, LTFT:  $20 \pm 4$  °C, HTCT: 25 °C, HTFT:  $25 \pm 4$  °C. The error bars show the standard deviations (n=3) and the letters above the bars denote the statistically significant differences (p<0.05).

Table 2: The elemental molar ratios of C: N, N: P, C: P, C: BSi (mol:mol) and quality ratio of C: Chl a (g:g) in the four experimental treatments of *Thalassiosira* sp. and *N. closterium f. minutissima*. Superscript letters represent significant differences (p<0.05).

|                            | 20 °C                |                             | 25 ℃                       |                         |  |
|----------------------------|----------------------|-----------------------------|----------------------------|-------------------------|--|
|                            | ± 0 °C               | ± 4 °C                      | ± 0 °C                     | ± 4 °C                  |  |
|                            | (LTCT)               | (LTFT)                      | (HTCT)                     | (HTFT)                  |  |
| Thalassiosira sp.          |                      |                             |                            |                         |  |
| C: N                       | $5.73\pm0.83$ a      | $7.50 \pm 0.66$ a           | $6.17 \pm 0.93$ a          | $1.03 \pm 0.00$ b       |  |
| N: P                       | $9.62 \pm 1.96$ a    | $7.71\pm0.27^{\rm \ a}$     | $6.20\pm2.03^{\mathrm{a}}$ | $31.86 \pm 0.88$ b      |  |
| C: P                       | $54.08 \pm 4.31^a$   | 57.92±6.99 a                | $43.79 \pm 1.33$ a         | $32.97 \pm 0.90$ b      |  |
| C: BSi                     | $4.20 \pm 0.15^{a}$  | $4.46 \pm 0.14^{\rm \ a}$   | $4.23\pm0.32^{\;a}$        | $3.74\pm0.65$ a         |  |
| C: Chl a                   | $57.53 \pm 2.75$ a   | $60.22 \pm 0.56$ a          | $51.22 \pm 4.34^{a}$       | $33.01 \pm 2.62^{b}$    |  |
| N. closterium f. minutissi | ima                  |                             |                            |                         |  |
| C: N                       | $5.92 \pm 0.53$ a    | $5.41\pm1.12^{a}$           | $5.72\pm0.78$ a            | $1.08 \pm 0.00$ b       |  |
| N: P                       | $13.07 \pm 0.94$ a   | 15.53±3.32 a                | $14.56 \pm 2.92$ a         | $47.07 \pm 1.56$ b      |  |
| C: P                       | $77.11 \pm 3.37^{a}$ | $81.53 \pm 1.92$ a          | $81.76 \pm 5.07^{a}$       | $50.68 \pm 1.79$ b      |  |
| C: BSi                     | $19.11 \pm 0.25$ a   | $20.82 \pm 1.72^{\ a}$      | $21.69 \pm 0.43$ a         | $20.01\pm3.62^{a}$      |  |
| C: Chl a                   | $73.57 \pm 1.13$ a   | $70.70\pm1.62^{\mathrm{a}}$ | 71.47±2.21 a               | 37.44±1.36 <sup>b</sup> |  |

# 4.4 Particulate organic carbon productivity and sinking rate

No significant changes in the POC productivities of *Thalassiosira* sp. and *N. closterium* f. minutissima were observed under HTCT compared to LTCT (Fig. 5A, B). In contrast, the POC productivity of *Thalassiosira* sp. decreased significantly by 42.44 % from 27.41± 6.09 to 15.78 ± 2.02, under HTFT compared to HTCT, while the POC productivity of *N. closterium* f. minutissima exhibited a non-significant increase (Fig. 5A, B).

The POC productivities of *Thalassiosira* sp. and *N. closterium* f. minutissima responded to temerapture fluctuation under low temperature (20 °C), but showed opposite reponses to temperature fluctuation under elevated temperature of (25 °C) (Fig. 5A). At LTFT, the POC productivities of *Thalassiosira* sp. and *N. closterium* f. minutissima were 35.88 ± 4.27 and 5.73 ± 0.57, increased by 60.98 % and 59.49 % compared to LTCT, respectively (p

Figure 5: The POC productivity (A, B) and sinking rate (C, D) of *Thalassiosira sp.* (A, B) and *N. closterium* f. minutissima (C, D) grown under LTCT:  $20 \,^{\circ}$ C, LTFT:  $20 \pm 4 \,^{\circ}$ C, HTCT:  $25 \,^{\circ}$ C, HTFT:  $25 \pm 4 \,^{\circ}$ C. The error bars show the standard deviations (n=3) and the letters above the bars denote the statistically significant differences.

# 4.5 Interactive effects of warming and temperature fluctuations

Two-way ANOVA revealed that warming had significant effects on most physiological parameters of *Thalassiosira* sp., including growth rate, Chl *a* content, protein, carbohydrate contents, PON, POC productivity, sinking rate, and C: N, C: P, N: P and C: Chl *a* ratios (Table 3). Temperature fluctuation had significant effects on the content of protein and PON, sinking rate, and C: N, N: P, C: Chl *a* ratios (Table 3). Significant interactive effects between warming and temperature fluctuation were observed for all the examined physiological parameters of *Thalassiosira* sp. except for cellular BSi content (Table 3). For *N. closterium* f. minutissima, both warming and temperature fluctuations had significant effects on all physiological parameters, except for C: BSi ratio (Table 3). Warming and temperature fluctuations produced significant interactive effects on the growth rate, cellular Chl *a*, protein, PON, and POP contents, as well as C: N, C: P, N: P, and C: Chl *a* ratios of *N. closterium* f. minutissima (Table 3).

Warming and temperature fluctuations had significant antagonistic interactive effects on the growth rates of *Thalassiosira* sp. and *N. closterium* f. minutissima but showed significant synergistic interactions on cellular protein and PON content as well as C: N, C: P, N: P and C: Chl a ratios (Table 3). For *Thalassiosira* sp., warming and temperature fluctuation had significant antagonistic interactions on carbohydrate, BSi content, POC productivity, and sinking rate, whereas no significant interactive effects were found on these parameters in *N. closterium* f. minutissima (Table 3). For elemental ratios, significant synergistic interactions between warming and temperature fluctuation were found in both *Thalassiosira* sp. and *N. closterium* f. minutissima, with no significant interactive effects on C: BSi ratio in either species (Table 3). In addition, warming and temperature fluctuation produced significant antagonistic interactive effects on the POP content of *Thalassiosira* sp., while in *N. closterium* f. minutissima, significant synergistic interactions were observed on POP content (Table 3).

Table 3: Interactive effects of warming and temperature fluctuations on the physiological parameters of *Thalassiosira* sp. and *N. closterium* f.minutissima. "-" represents antagonistic effects and "+" represents synergistic effects. "\*" represents significance and "ns" represents non-significance (two-way ANOVA, p=0.05).

|                  | Thalassiosira sp. |    |             |             | Nitzschia closterium f. minutissima |    |             |             |
|------------------|-------------------|----|-------------|-------------|-------------------------------------|----|-------------|-------------|
|                  | Two-way ANOVA     |    |             | Type of     | Two-way ANOVA                       |    |             | Type of     |
|                  | Warming           | FT | Interaction | interaction | Warming                             | FT | Interaction | interaction |
| Growth rate      | *                 | ns | *           | -           | *                                   | *  | *           | -           |
| Chl a            | *                 | ns | *           | -           | *                                   | *  | *           | +           |
| Protein          | *                 | *  | *           | +           | *                                   | *  | *           | +           |
| Carbohydrate     | *                 | ns | *           | -           | *                                   | *  | ns          | -           |
| POC              | ns                | ns | *           | -           | *                                   | *  | ns          | -           |
| PON              | *                 | *  | *           | +           | *                                   | *  | *           | +           |
| POP              | ns                | ns | *           | -           | *                                   | *  | *           | +           |
| BSi              | ns                | ns | *           | -           | *                                   | *  | ns          | +           |
| POC productivity | *                 | ns | *           | -           | *                                   | *  | ns          | -           |
| Sinking rate     | *                 | *  | *           | -           | *                                   | *  | ns          | -           |
| C: N             | *                 | *  | *           | +           | *                                   | *  | *           | +           |
| C: P             | *                 | ns | *           | +           | *                                   | *  | *           | +           |
| N: P             | *                 | *  | *           | +           | *                                   | *  | *           | +           |
| C: BSi           | ns                | ns | ns          | +           | ns                                  | ns | ns          | -           |
| C: Chl a         | *                 | *  | *           | +           | *                                   | *  | *           | +           |

# 5 Discussion

Our experimental results demonstrate that both *Thalassiosira* sp. and *N. closterium* f. minutissima exhibit enhanced thermal tolerance under fluctuating temperature regimes compared to constant temperature conditions. Our findings also reveal significant species-specific variability in physiological responses to warming and temperature fluctuations, with the pennate diatom *N. closterium* f. minutissima demonstrating greater resilience to elevated temperatures than *Thalassiosira* sp.. Particularly noteworthy are the differential responses observed in POC productivity and sinking rates between the two species under varying thermal regimes, which may alter their respective contributions to carbon export. These results not only emphasize the critical importance of incorporating natural temperature variability but also highlight the necessity of considering species-specific physiological responses when modeling and predicting the ecological impacts of climate change on marine ecosystems.

Figure 6: Schematic diagram of the responses of *Thalassiosira* sp. and *N. closterium* f. minutissima to warming under constant temperature and fluctuating temperature. Arrow thickness represents the magnitude of change, with red arrows indicating significant increases, blue arrows indicating significant decreases, and horizontal lines denoting no significant changes.

#### 5.1 Temperature fluctuation alleviated the negative effects of warming on growth

When the temperature was raised from 20 °C to 25 °C in our investigation, the growth rates of *Thalassiosira* sp. and *N. closterium* f. minutissima both dramatically dropped (Fig. 1, constant temperature treatments). Similarly, previous study on marine diatom *Thalassiosira weissflogii* suggest an optimal temperature of 20 °C for the growth of *T. weissflogii* (Taucher et al., 2015). The optimal temperature of marine diatom *N. closterium* f. minutissima is 22 °C. Therefore, here we assume that 25 °C of our high temperature treatments has exceeded the optimal temperature of both species. In general, below the optimal temperature, rising temperature enhances enzyme activity, which in turn stimulates cellular metabolism and promotes growth (Arrhenius, 1889). At higher temperatures above the optimum, the inhibition on the growth occurs due to cellular stress (Cai et al., 2022; Edullantes et al., 2023).

However, in our investigation, temperature variation had distinct effects on growth at low and high temperatures. Temperature variation had no discernible effect on either species' development rate at 20 °C (Fig. 2, Table 2). This tolerance may reflect their coastal habitat, where phytoplankton cells are subjected to frequent temperature fluctuations (annual temperature range between 15 °C and 30 °C, daily temperature increases can reach up to 5 °C). Phytoplankton in these dynamic environments may have developed mechanisms to cope with environmental heterogeneity, maintaining stable growth even under fluctuating conditions. Our results are consistent with models that predict enhanced growth under non-optimal temperature fluctuations, especially in species adapted to variable coastal environments (Bernhardt et al., 2018).

Interestingly, our study observed that temperature fluctuation increased the resilience of the growth of both diatom species to warming. At high temperature (25 °C), temperature fluctuation mitigated the negative effects on growth by warming.

Especially for *N. closterium* f. minutissima, the growth rate of HTFT treatment showed no significant difference from that at low temperature (Fig. 2). This indicates that temperature fluctuations may intermittently expose cells to more favorable conditions during the cooling phase (Wolf et al., 2024; Schaum et al., 2018), facilitating recovery from stress. Similarly, previous study on *Thalassiosira pseudonana* demonstrated that temperature fluctuation (cycling between 22 °C and 32 °C) facilitate rapid cellular adaptation by modulating transcription and oxidative stress responses (Schaum et al., 2018).

Therefore, temperature fluctuation in natural environments play a critical role in driving adaptation processes. The significant antagonistic interactions of warming and temperature fluctuation on growth rates of both diatom species (Table 2) suggested that temperature fluctuations might alleviate the stress caused by extreme temperatures, enhancing the resilience of marine diatoms under warming scenarios.

# 5.2 Elemental composition and resource allocation

Temperature significantly affects phytoplankton metabolism and the utilization of nutrients within the cells, which in turn affects the elemental composition of phytoplankton (Toseland et al., 2013; Spilling et al., 2015). In our study, warming (form 20 °C to 25 °C) at constant temperature led to significantly increase in cellular contents of POC, POP, and BSi in *Thalassiosira* sp., as well as significantly elevated POC, PON, and POP quotas in *N. closterium* f.minutissima (Fig. 4). Whereas the cellular POC, POP and BSi contents of a temperate diatom species *Thalassiosira* sp. decreased with warming (form 15 °C to 20 °C) in previous study (Cai et al., 2022), this difference may be due to the fact that the temperatures of the experimental setups were on different sides of the thermal response curve of *Thalassiosira* sp.

Both species showed elevated cellular protein and PON contents at HTFT compared to LTFT, indicating a reallocation of nitrogen to protein synthesis. This adaptive mechanism presumably facilitated the upregulation of stress-responsive proteins, thereby enhancing the capacity of the cells to cope with environmental stressors, such as extreme temperatures and thermal fluctuation (O'donnell et al., 2018). For the larger-celled *Thalassiosira* sp., the higher metabolic demand for protein synthesis led to a reduction in the cellular quotas of macromolecules, including carbohydrates, POC, and BSi at HTFT compared to LTFT (Fig. 3, 4). In comparison, *N. closterium* f.minutissima exhibited similar responses to temperature fluctuation at low and high temperatures. The increased protein, PON, carbohydrate, and POP content at HTFT compared to LTFT likely facilitated recovery of growth rates, highlighting the higher tolerance of *N. closterium* to temperature fluctuations and thermal stress (Fig. 3, 4).

Phytoplankton exhibit nonlinear responses to temperature, and thus temperature fluctuation influence their elemental composition in distinct ways (Wang et al., 2019). These variations drive changes in cellular resource allocation, which in turn impact their stoichiometry and ultimately shape the competitive dynamics among different species (O'donnell et al., 2018; Baker et al., 2016). Despite these changes in elemental content, we observed warming had no significant effects on elemental ratios at constant temperature (Table 1). Similarly, no significant effects of warming on C: N and N: P were

observed by previous study on the Antarctic diatoms *Pseudo-nitzschia subcurvata* and *Chaetoceros* sp. (Zhu et al., 2016). In our study, HTCT did not significantly affect the elemental ratios of either species compared to LTCT. However, at HTFT, the C: N, C: P and C: Chl *a* ratios significantly reduced in both species, while the N: P ratio increased compared to LTFT (Table 1).

## 370 5.3 Organic carbon fixation and sinking rate – oceanic implications

Marine diatoms are key primary producers in the ocean (Field et al., 1998). Due to the ballast effect of their silica frustules and their ability to produce POC, studying their POC production and sinking rate in response to warming and temperature fluctuations is crucial for understanding their role in the ocean's carbon cycle.

For Thalassiosira sp., HTCT had no significant effect on POC productivity compared to LTCT (Fig. 5A), likely because warming under constant temperature reduced growth rate but increased cellular POC content, offsetting the decline in POC productivity caused by slower growth. In contrast, the sinking rate of *Thalassiosira* sp. significantly increased under HTCT compared to LTCT (Fig. 5C), potentially due to elevated cellular POC and BSi content (Fig. 4). However, warming under temperature fluctuation elicited different responses. The POC productivity of *Thalassiosira* sp. significantly decreased under HTFT compared to LTFT (Fig. 5A), attributed to unchanged growth rates but reduced cellular POC content (Fig. 2, 4). Similarly, HTFT significantly lowered the sinking rate of *Thalassiosira* sp. compared to LTFT (Fig. 5C), likely driven by declines in cellular POC and BSi content (Fig. 4). Thus, warming under constant temperature and temperature fluctuation had distinct impacts on *Thalassiosira* sp. At low temperature, temperature fluctuation enhanced POC productivity (Fig. 5A), supported by stable growth rates and increased POC content (Fig. 2A, 4A). At high temperature, despite accelerated growth rates under temperature fluctuation (Fig. 2A), POC productivity significantly declined due to reduced POC content (Fig. 4A). For N. closterium f. minutissima, warming under both constant and fluctuating temperature had no significant effect on POC productivity or sinking rate (Fig. 5B, D). This further indicates that the smaller-celled N. closterium f. minutissima exhibits a superior ability to balance resource acquisition and utilization at elevated temperatures, conferring greater resilience to warming (Fan et al., 2023). A comparative study on diatoms of varying sizes confirmed that cell size decreases with rising temperatures, with larger species like *Thalassiosira punctigera* showing greater vulnerability to thermal stress, while smaller species exhibit enhanced tolerance to warming (Fan et al., 2023). However, temperature fluctuation increased POC productivity at both low and high temperatures (Fig. 5). At low temperature, this was due to increased POC content and stable growth rate, while at high temperature, it resulted from stable POC content and accelerated growth rate (Fig. 2, 4). For N. closterium f. minutissima, temperature fluctuation at low temperature elevated POC content and sinking rates, although BSi content remained unaffected (Fig. 4). Previous studies indicated that the sinking rate of Thalassiosira sp. is approximately five times higher than that of N. closterium f. minutissima, consistent with the general observation that larger diatoms exhibit higher sinking rates and contribute more significantly to deep-sea carbon export (Cai et al., 2022; Kiørboe, 1993). Thus, *Thalassiosira* sp. is expected to play a more prominent role in carbon export flux under warming conditions.

In summary, temperature fluctuation exerted contrasting effects on the carbon export flux of *Thalassiosira* sp. at different temperature regimes. At low temperature, it increased POC, BSi content, and sinking rates (Fig. 4, 5), thereby enhancing carbon export. Conversely, at high temperature, it reduced POC, BSi content, and sinking rates (Fig. 4, 5), leading to diminished carbon export to the deep ocean. These findings underscore the importance of incorporating temperature fluctuation into studies of phytoplankton and marine carbon dynamics.

# 5.4 Differential thermal responses driven by species traits

The differential physiological responses of *Thalassiosira* sp. and *Nitzschia closterium* f. minutissima to warming and temperature fluctuations are likely attributable to inherent differences in their morphology and ecological niches. Generally, algal cellular utilization of both light energy and nutrients, as well as metabolic efficiency, are intrinsically associated with cell size (Marañón, 2015; Marañón et al., 2012). As a representative centric diatom, *Thalassiosira* sp. typically has a larger cell size (~30 μm), leading to fast sinking into depth but also impose higher metabolic costs under thermal stress. Conversely, the smaller pennate diatom *N. closterium* f. minutissima (~15 μm) exhibits a higher surface-area-to-volume ratio, promoting more efficient nutrient uptake and gas exchange, especially in variable environmental conditions. Additionally, pennate diatoms are commonly found in benthic or nearshore habitats that experience greater environmental heterogeneity (Burden et al., 2020), thus with increased adaptability and thermal resilience to temperature fluctuations observed in the present study. These morphological and geographical differences likely underpin species-specific strategies to thermal tolerance, and the consequent resource allocation and carbon export, highlighting the necessity of incorporating taxonomic and functional diversity when evaluating phytoplankton responses to climate change.

## **6 Conclusion**

Our findings demonstrated that temperature fluctuations enhance the thermal tolerance of diatoms, though the degree of adaptation varies between species. The large-celled *Thalassiosira* sp. redirected resources toward protein synthesis to cope with thermal stress, while the small-celled *N. closterium* f. minutissima exhibited higher sensitivity to temperature fluctuations and a greater capacity for cellular repair. The divergent responses in carbon fixation and sinking rates further underscore species-specific contributions to oceanic carbon fluxes. These findings further provide insight into understanding how future climate conditions may alter phytoplankton productivity and biogeochemical cycling in marine ecosystems. Future studies should also investigate the combined effects of multiple stressors, such as nutrient availability and light conditions, with temperature fluctuations, as well as long-term adaptive responses of marine phytoplankton to environmental heterogeneity.

Data availability. The research data are available at https://doi.org/10.5281/zenodo.15274949 (Sheng et al., 2025).

- Author contributions. Conceptualization: YW, YF; Methodology: YW, YF; Investigation: YW, FY; Supervision: YF, YW, 430 AF, SBI; Writing—original draft: YS; Writing—review & editing: YS, YF, AF, SBI.
  - *Competing interests*. The authors declare that the research was conducted in the absence of any commercial or financial relationships that could be construed as a potential conflict of interest.
- *Disclaimer*. Publisher's note: Copernicus Publications remains neutral with regard to jurisdictional claims made in the text, published maps, institutional affiliations, or any other geographical representation in this paper. While Copernicus Publications makes every effort to include appropriate place names, the final responsibility lies with the authors.
- Acknowledgements. The authors would like to thank Dr. Xinwei Wang and Prof. Haibo Jiang's Laboratory at the Ningbo University for helping with the analysis of the elemental compositions, Dr. Ruoyu Guo at the Second Institute of Oceanography, Ministry of Natural Resources for providing phytoplankton stock cultures, Dr. Guisheng Song at Tianjin University for helping with the Chl a analysis, and Shanghai Frontiers Science Center of Polar Science, Shanghai Jiao Tong University, 1954 Huashan Road, Shanghai 200030, China.
- *Financial support.* This work was financially supported by the National Natural Science Foundation of China grant (42276093), the Oceanic Interdisciplinary Program of Shanghai Jiao Tong University (SL2022PT203).

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
