# Peer review of "Temperature fluctuation alleviates the negative effects of warming on marine diatoms: comparison between *Thalassiosira* sp. and *Nitzschia closterium* f. minutissima"

_EGUsphere, 2025_

## Author Comment (AC1)

**RC1:**

Comments on "Temperature fluctuation alleviates the negative effects of warming on marine diatoms: comparison between *Thalassiosira* sp. and *Nitzschia closterium* f. minutissima" by Sheng et al.

**General Comments:** This manuscript addresses a compelling and timely topic by exploring how temperature fluctuations influence the physiological and biogeochemical responses of marine diatoms to ocean warming, an aspect often overlooked in studies conducted under static temperature conditions. The authors provide valuable data on two ecologically significant diatom species, *Thalassiosira* sp. and *Nitzschia closterium* f. minutissima, revealing species-specific responses in growth rate, particulate organic carbon (POC), biogenic silica (BSi), and sinking rate. These findings contribute to our understanding of how diatom-driven biogeochemical cycles may respond to future ocean conditions. The manuscript is well-structured, clearly written, and supported by robust experimental methods. However, I have several minor concerns and suggestions to enhance the manuscript's clarity, scientific rigor, and overall impact.

**Response:**

We sincerely appreciate the reviewer's thoughtful evaluation of our manuscript and their constructive suggestions, which have helped us improve the study's clarity and scientific rigor. Below, we address the reviewer's specific comments point by point.

**Specific Comments:**

**Lines 92–94:** Please provide details on the season when *Thalassiosira* sp. and *Nitzschia closterium* f. minutissima were isolated from the Yellow Sea, along with the corresponding mean water temperature at the time of collection. Additionally, clarify how long these species were maintained in the laboratory prior to the experiments, as this could influence their acclimation to culture conditions.

**Response:**

Line 98-102: The marine diatoms *Thalassiosira* sp. and *Nitzschia closterium* f. minutissima were originally isolated from surface seawater (depth 3 – 12 m; salinity 34.78) at 118° 58.055′E, 38°39.111′N (China) in August 2019. At the time of sampling,

the sea surface temperature was approximately 15 °C. Stock cultures of two diatoms were maintained in sterilized f/2 medium natural seawater, in Nalgene polycarbonate bottles in a constant temperature and illumination incubator (GXZ-280D, NingBo) at 20 °C.

**Line 102:** Correct the typographical error in the cell abundance notation. The number "4" in "1 × 104 cells mL-1" should be formatted as a superscript (i.e., 1 × 10⁴ cells mL⁻¹).

**Response:**

We have made the correction in the manuscript.

**Revision:**

Line 113: For each temperature treatment, stock cultures in the logarithmic growth phase were inoculated into triplicate polycarbonate culture flasks (1L, Nalgene, USA), with an initial cell abundance of $1\times10^4$ cells mL$^{-1}$. The light intensity was maintained at 170-200 µmol photons m$^{-2}$ s$^{-1}$ with light: dark cycle of 12 h: 12 h.

**Line 110:** Clarify whether Figure 1 represents recorded temperature changes from the experiment or is a schematic diagram of the temperature treatments. If it is a schematic, consider including a supplementary figure with actual temperature data to validate the experimental setup.

**Response:**

Figure 1 is a schematic diagram illustrating the temperature treatments used in this experiment. We did monitor the temperature in the water baths using water temperature data loggers (MX2201, HOBO, USA). The measured temperature was within a range of ± 0.2 °C from the preset temperature.

In the revised manuscript, we have clarified in the figure legend that Figure 1 is a schematic diagram.

**Revision:**

Line 111: The measured temperature was within a range of ± 0.2 °C from the preset temperature.

Line 121: Figure 1: Temperature schematic diagram. LTCT: 20 °C, LTFT: 20 ± 4 °C, HTCT: 25 °C, HTFT: 25 ± 4 °C.

**Line 115:** Specify the storage conditions for samples during chlorophyll *a* (Chl *a*) extraction. Were the samples stored in the dark, and at what temperature (e.g., 4°C)?

**Response:**

We have added the detailed information for sample storage conditions.

**Revision:**

Line 128: For chlorophyll *a* (Chl *a*) measurement, 20 mL sample was filtered onto GF/F filters (Whatman, USA). After extraction with 5 mL of 90 % acetone for 24 hours (stored at -20 °C in the dark), the Chl *a* content was measured using a fluorometer (Trilogy, Turner Designs, USA).

**Line 118:** For the growth rate calculations, confirm whether Chl *a* fluorescence was measured directly from the algal culture or after filtration. If measured directly, discuss any potential interference from culture medium or cell aggregation that might affect fluorescence readings.

**Response:**

For the growth rate calculations, the *in vivo* Chlorophyll *a* fluorescence was measured directly from the algal cultures without filtration. For the *in vivo* fluorescence measurements, the culture medium was always used as blank, therefore the potential interference from culture medium was eliminated. In addition, we conducted a semi-continuous incubation with cells growing at the exponential phase, and the cultures in the incubation bottles were well mixed for a 2-3 times a day, as such, there was no cell aggregation happening in the culture system.

**Revision:**

Line 130: 2 mL sample was kept in the dark for 20 minutes, followed by measurement of the in vivo Chl *a* fluorescence using a fluorometer (Trilogy, Turner Designs, USA). For the in vivo fluorescence measurements, the culture medium was always used as blank, therefore the potential interference from culture medium was eliminated. In addition, we conducted a semi-continuous incubation with cells growing at the exponential phase, and the cultures in the incubation bottles were well mixed for a 2-3 times a day, as such, there was no cell aggregation happening in the culture system.

**Line 158:** Origin 2021 is a product of OriginLab Corporation, not "Tukey", which is a statistical method.

**Response:**

We have made the correction as suggested.

**Revision:**

Line 175: Significance analysis and interaction effects were performed by two-way ANOVA using Origin 2021 software (OriginLab Corporation, USA). Differences between treatments were considered significant at level of p < 0.05. The pairwise tests between treatments were conducted using Tukey's multiple comparison post-hoc analysis.

Discussion Section: I recommend adding a paragraph to discuss the differential responses between *Thalassiosira* sp. (a centric diatom) and *Nitzschia closterium* f. minutissima (a pennate diatom). Highlight how their morphological and ecological differences (e.g., cell size, silica structure, or habitat preferences) might contribute to their distinct responses to warming and temperature fluctuations. This would strengthen the manuscript's ecological and taxonomic insights.

**Response:**

We agree that adding a paragraph to compare the differential responses between the two species will strengthen the ecological implications of our study. Now a new paragraph has been incorporated into the Discussion to clarify the differences between *Thalassiosira* sp. (a centric diatom) and *Nitzschia closterium* f. minutissima (a pennate diatom).

**Revision:**

Line 401:

**5.4 Differential thermal responses driven by species traits**

The differential physiological responses of *Thalassiosira* sp. and *Nitzschia closterium* f. minutissima to warming and temperature fluctuations are likely attributable to inherent differences in their morphology and ecological niches. Generally, algal cellular utilization of both light energy and nutrients, as well as metabolic efficiency, are intrinsically associated with cell size (Marañón, 2015; Marañón et al., 2012). As a representative centric diatom, *Thalassiosira* sp. typically has a larger cell size (~30 μm), leading to fast sinking into depth but also impose higher metabolic costs under thermal

stress. Conversely, the smaller pennate diatom *N. closterium* f. minutissima (~15 μm) exhibits a higher surface-area-to-volume ratio., promoting more efficient nutrient uptake and gas exchange, especially in variable environmental conditions. Additionally, pennate diatoms are commonly found in benthic or nearshore habitats that experience greater environmental heterogeneity (Burden et al., 2020), thus with increased adaptability and thermal resilience to temperature fluctuations observed in the present study. These morphological and geographical differences likely underpin species-specific strategies to thermal tolerance, and the consequent resource allocation and carbon export, highlighting the necessity of incorporating taxonomic and functional diversity when evaluating phytoplankton responses to climate change.

**References cited:**

Burden, A., Smeaton, C., Angus, S., Garbutt, A., Jones, L., Lewis, H., and Rees, S.: Impacts of climate change on coastal habitats, relevant to the coastal and marine environment around the UK., MCCIP Sci. Rev., pp:228-255, https://doi.org/10.14465/2020.arc11.chb, 2020.

Marañón, E.: Cell Size as a Key Determinant of Phytoplankton Metabolism and Community Structure. Annu. Rev. Mar. Sci., 7, 241-264, https://doi.org/10.1146/annurev-marine-010814-015955, 2015.

Marañón, E., Cermeño, P., López-Sandoval, D. C., Rodríguez-Ramos, T., Sobrino, C., Huete-Ortega, M., Blanco, J. M., Rodríguez, J., Fussmann, G.: Unimodal size scaling of phytoplankton growth and the size dependence of nutrient uptake and use. Ecol. Lett., 16, 371-379, https://doi.org/doi:10.1111/ele.12052,2012.

---

## Author Comment (AC2)

**RC2:**

In this paper, Sheng et al. investigated the response of two marine diatom species, *Thalassiosira* sp. and *Nitzschia closterium* f. minutissima, to ocean warming and temperature fluctuation ($\pm 4$ °C) under low (20 °C) and high (25 °C) average temperatures. The semi-continuous incubation method was adopted. Their results demonstrate that temperature fluctuation alleviated the negative effects of elevated temperatures on the growth of both species and revealed distinct responses of the two diatoms in cellular element contents and sinking rate. This study explored the influence of temperature fluctuation on the physiology of marine diatoms and shed light on the biogeochemical feedbacks in the context of global warming.

In general, the methods and the analyses are very sound, and the interpretation of the results are overall appropriate. Moreover, the manuscript is generally well-written and referenced. I feel that this is in principle an excellent study. However, several points need to be addressed before acceptance.

**Response:**

We sincerely appreciate the reviewer's thoughtful evaluation of our manuscript and constructive suggestions, which have helped us improve the overall quality of the manuscript. Below, we address the reviewer's specific comments point by point.

**Comments:**

**Line 25-30:** "However, warming significantly decreased the cellular particulate organic carbon (POC) and biogenic silica (BSi) contents, and sinking rate, while increasing protein content to cope with the thermal stress under temperature fluctuation. Temperature fluctuation at low average temperatures significantly increased the cellular POC and BSi contents, as well as POC productivity and sinking rate, while at high average temperatures, these parameters were significantly decreased." The two sentences are somewhat repetitive, which may cause confusion. Please revise for conciseness.

**Response:**

We have revised the sentence for clarity and conciseness.

**Revision:**

Line 25-29: However, warming significantly reduced cellular particulate organic carbon (POC) and biogenic silica (BSi) contents, as well as sinking rates, while increasing the protein content to cope with the thermal stress under temperature fluctuation. The effects of temperature fluctuation were dependant on the average temperature: at low average temperature, temperature fluctuation increased cellular POC, BSi, POC productivity, and sinking rates, whereas at high average temperature, these parameters were decreased significantly.

**Line 40:** Please clarify the time span over which the 1°C increase is expected to occur or has already occurred.

**Response:**

We have added the time span in the revised manuscript.

**Revision:**

Line 40: This will in turn induce global warming, with an average increase in global temperature by approximately 1 °C above 1850–1900 in 2011–2020.

**Line 53:** It would not be appropriate to use "detrimental" here since the following sentence highlighted "the temperature fluctuation (2 days) reduced the mortality rate of *Emiliania huxleyi*". Please consider using a different word to describe the complex situation. Also, please add basic descriptions for the coccolithophore *Emiliania huxleyi*, in case that readers are not familiar with it.

**Response:**

Thanks for pointing out the problem. We have made the following revision in the manuscript.

**Revision:**

Line 55-59: Ocean warming not only increases the average ocean temperature but also enhances the frequency and intensity of temperature fluctuation, which may have

complex effects on marine organisms than warming alone (Ketola and Saarinen, 2015). It has been reported that high-frequency temperature fluctuation (2 days) reduced the mortality rate of the coccolithophore *Emiliania huxleyi*, a dominant species of the calcifying phytoplankton group that play a key role in the calcium carbonate production and the marine carbon cycling (Wang et al., 2019).

**Line 86:** Please add more details explaining why the two diatoms were selected, e.g. their ecological roles, importance, size, common and different characteristics. Some contents in Line 300-310 could be moved to the Introduction.

**Response:**

Thank you for the suggestion. We have added some detailed information of the two diatom species selected for the experiment. In addition, we have also moved some of the more general contents in Line 300-310 to the Introduction section (Line 44-46).

**Revision:**

Line 91-93: Two representative diatom species with distinct cell sizes were selected: the centric diatom *Thalassiosira* sp. and the pennate diatom *Nitzschia closterium* f. minutissima. Both species are model diatoms, representing the two major taxonomic class of diatoms (Centric vs. Pennate), allowing for comparison of responses to environmental changes between two distinct diatom groups.

**Line 95:** Please provide more information for the experiments. Why "20°C" and "25°C" were chosen as the average temperature for "cold" and "warm" status? Have the optimal growth temperatures of two diatoms been determined using the temperature curves? How many days have been taken to reach the stabilized growth rate? Unless this experiments took years to complete, please use the words like "adapt" or "adaptation" with caution. "Acclimation" would be a better word for the short-term experiments.

**Response:**

The selection of 20 °C and 25 °C was based on both ecological relevance and experimental design considerations. We have previously obtained the thermal response curves of the two species, the optimal growth temperature for *Thalassiosira* sp. was ~

19 °C, and ~ 22 °C for *Nitzschia closterium* f. minutissima. We have incorporated the above content into the revised manuscript (Line 106-108).

We chose 20 °C because it is close to the optimal temperature for both species, which also was the stock culture growing temperature. 25 °C represents a moderately elevated temperature condition above the optimal range for both species, simulating a typical scenario with 5 °C of warming projected for the end of the century (IPCC, 2023). This allows us to examine the physiological responses of both diatoms to a temperature shift from near-optimal to supra-optimal conditions.

We defined stable growth as a condition in which the daily growth rate varied by less than 10 % over five consecutive days, and the total acclimation time for both species was 18 days.

**References cited:**

IPCC: 2023: Summary for Policymakers. In: Climate Change 2023: Synthesis Report. Contribution of Working Groups I, II and III to the Sixth Assessment Report of the Intergovernmental Panel on Climate Change [Core Writing Team, H. Lee and J. Romero (eds.)]. IPCC, Geneva, Switzerland, pp. 1-34, https://doi.org/10.59327/IPCC/AR6-9789291691647.001., 2023.

Line 96: Please consider including a table of abbreviations like "LTCT", "LTFT","HTCT" and "HTFT".

**Response:**

Although the abbreviations "LTCT," "LTFT," "HTCT," and "HTFT" were defined in the section 2.2, we agree that including a table of abbreviations will improve clarity and readability. We have now added a table listing relevant abbreviations in the revised manuscript.

**Revision:**

Line 123:

**Table 1:Abbreviation table of temperature treatment group**

| Abbreviation | Full term | Temperature treatment |
| --- | --- | --- |
| LTCT | Low and constant temperature | 20 °C |
| LTFT | Low and fluctuation temperature | 20 ± 4°C |
| HTCT | High and constant temperature | 25 °C |

| HTFT | High and fluctuation temperature | $25 \pm 4°C$ |
| --- | --- | --- |

**Line 163-165:** Please add more details for the formula and rephrase the sentence "When the positive and negative results of $OE_1$ and $OE_2$ are the same, ⋯and vice versa, they are antagonistic interaction effects". Are the results of $OE_1$ and $OE_2$ always be opposite?

**Response:**

We appreciate the comment and have revised the text to improve clarity.

**Revision:**

Line 182-184: When $|OE_{1+2}| > |ME_{1+2}|$, the interaction between the two environmental factors is synergistic, whereas when $|OE_{1+2}| < |ME_{1+2}|$, the interaction becomes antagonistic.

**Line 313:** It would be a good idea to add the information about the temperature range and fluctuation conditions in the coastal habits where the two diatoms were collected, to interpret the results in the context of evolution.

**Response:**

We fully agree that incorporating information on the temperature range and fluctuations in the coastal habitats where the diatoms were collected would help interpret the results more comprehensively from an evolutionary perspective. Based on temperature tracker (https://www.marineheatwaves.org/), the annual temperature fluctuation in the isolation area ranges approximately from 15 °C to 30 °C. Studies conducted in this region have also reported that daily temperature increases can reach up to 5 °C. We have incorporated this information into the revised manuscript (Line 315).

**Revision:**

Line 327-329: This tolerance may reflect their coastal habitat, where phytoplankton cells are subjected to frequent temperature fluctuations (annual temperature range between 15 °C and 30 °C, daily temperature increases can reach up to 5 °C).

Although species-specific responses to environmental factors are widely acknowledged, it would be helpful if the authors could interpret the different responses of two diatoms revealed here based on their distinct characteristics, and add this point to the Discussion section.

**Response:**

We appreciate the reviewer's valuable suggestion. A new paragraph has been incorporated into the Discussion to clarify the differences between *Thalassiosira* sp. (a centric diatom) and *Nitzschia closterium* f. minutissima (a pennate diatom).

**Revision:**

Line 401:

**5.4 Differential thermal responses driven by species traits**

The differential physiological responses of *Thalassiosira* sp. and *Nitzschia closterium* f. minutissima to warming and temperature fluctuations are likely attributable to inherent differences in their morphology and ecological niches. Generally, algal cellular utilization of both light energy and nutrients, as well as metabolic efficiency, are intrinsically associated with cell size (Marañón, 2015; Marañón et al., 2012). As a representative centric diatom, *Thalassiosira* sp. typically has a larger cell size (~30 μm), leading to fast sinking into depth but also impose higher metabolic costs under thermal stress. Conversely, the smaller pennate diatom *N. closterium* f. minutissima (~15 μm) exhibits a higher surface-area-to-volume ratio., promoting more efficient nutrient uptake and gas exchange, especially in variable environmental conditions. Additionally, pennate diatoms are commonly found in benthic or nearshore habitats that experience greater environmental heterogeneity (Burden et al., 2020), thus with increased adaptability and thermal resilience to temperature fluctuations observed in the present study. These morphological and geographical differences likely underpin species-specific strategies to thermal tolerance, and the consequent resource allocation and carbon export, highlighting the necessity of incorporating taxonomic and functional diversity when evaluating phytoplankton responses to climate change.

**References cited:**

Burden, A., Smeaton, C., Angus, S., Garbutt, A., Jones, L., Lewis, H., and Rees, S.: Impacts of climate change on coastal habitats, relevant to the coastal and marine environment around the UK., MCCIP Sci. Rev., pp:228-255, https://doi.org/10.14465/2020.arc11.chb, 2020.

Marañón, E.: Cell Size as a Key Determinant of Phytoplankton Metabolism and Community Structure. Annu. Rev. Mar. Sci., 7, 241-264, https://doi.org/10.1146/annurev-marine-010814-015955, 2015.

Marañón, E., Cermeño, P., López-Sandoval, D. C., Rodríguez-Ramos, T., Sobrino, C., Huete-Ortega, M., Blanco, J. M., Rodríguez, J., Fussmann, G.: Unimodal size scaling of phytoplankton growth and the size dependence of nutrient uptake and use. Ecol. Lett., 16, 371-379, https://doi.org/doi:10.1111/ele.12052,2012.

In the Discussion section, it's generally not necessary to cite figures again, especially if those figures are already presented and mentioned in the Results section.

**Response:**

We appreciate the suggestion and have removed the figure citations in the Discussion section.

There are several typos in the manuscript. For instance, there is a repeated use of 'overall' in Line 32 and Line 34, which could be streamlined. In Line 58, "Microcystis aeruginosa" should be italic. In Figure 6, the symbols of *Thalassiosira* in two columns are different. In Line 312, there is a space in the word "tolerance". Hence one more round of thorough proof reading would be in order.

**Response:**

Thank you for your detailed suggestions. We have thoroughly checked all the spellings throughout the manuscript and made the corrections.

**Revision:**

Line 34:  These findings highlight the important, yet often underestimated, influence of temperature fluctuation on the physiology of marine diatoms in the context of global warming, thus having implications for further understanding the biogeochemical feedbacks.

Line 56: Additionally, study on green algae has shown that temperature fluctuation slowed the growth rate of *Chlorella* and *Micromonas*, but did not affect the growth of *Microcystis aeruginosa* (Zhang et al., 2015).

[Figure]

[Figure]

**Figure 6: Schematic diagram of the responses of *Thalassiosira* sp. and *N. closterium* f. minutissima to warming under constant temperature and fluctuating temperature. Arrow thickness represents the magnitude of change, with red arrows indicating significant increases, blue arrows indicating significant decreases, and horizontal lines denoting no significant changes.**

Line 327: The error occurred during the Word-to-PDF conversion and has now been corrected as "tolerance".